A method for evaluating sediment-induced macroinvertebrate community composition changes in Idaho streams

http://orcid.org/0000-0002-0511-4416 Williams Jason 1 2 jason.williams@cadmusgroup.com
Efta James 3
1 Idaho Department of Environmental Quality , Lewiston, Idaho , United States
2 Cadmus Group , Waltham, MA , United States
3 Northern Region, United States Forest Service , Missoula, MT , United States
Silva Daniel
Electronic publication date: 2024 Oct 4
Publication date: 2024
Volume: 12
Electronic Location ID: e18060
Received 2024 Jun 18; Accepted 2024 Aug 17
Copyright: © 2024 Williams and Efta
Copyright year: 2024
Copyright holder: Williams and Efta
License: This is an open access article distributed under the terms of the Creative Commons Attribution License, which permits unrestricted use, distribution, reproduction and adaptation in any medium and for any purpose provided that it is properly attributed. For attribution, the original author(s), title, publication source (PeerJ) and either DOI or URL of the article must be cited.
License URL: https://creativecommons.org/licenses/by/4.0/

Keywords: Stream, Sediment, Macroinvertebrates, Clean water act, Idaho, Assessment

Funding: The authors received no funding for this work.

==============================
Human activities can increase sediment delivery to streams, changing the composition, distribution, and abundance of stream aquatic life. Few U.S. states have numeric water quality standards for streambed sediment under the Clean Water Act, so managers often need to develop local application-specific benchmarks. This study developed stream surface fine sediment <2 mm (sand and fines, SF) and macroinvertebrate fine sediment biotic index (FSBI) benchmarks and an application framework to test for sediment-induced macroinvertebrate community composition changes in 1st–4th order Idaho streams. FSBI reference benchmarks were calculated as the 25th percentile FSBI value among reference sites within three ecoregion-based site classes. Two approaches were used to develop SF benchmarks. Quantile regression was used to define reach-specific SF benchmarks representing an upper bound value expected under reference conditions. In addition, logistic regression was used to predict SF values with 50% and 75% probability that FSBI is worse than reference within each stream order and site class. The strength of association between SF benchmarks and macroinvertebrate community condition was evaluated by calculating relative risk using multiple datasets and examining responses of multiple macroinvertebrate indicators to SF benchmark status. SF reference benchmarks generally had stronger associations with poor macroinvertebrate condition than SF stressor-response benchmarks. Across datasets and macroinvertebrate indicators, poor macroinvertebrate condition was 1.8–3 times more likely when SF reference benchmarks were exceeded than when achieved. We propose rating the strength of evidence for a surface fine sediment-induced macroinvertebrate community composition change at the sample event scale as ‘unlikely’ if both SF and FSBI reference benchmarks are achieved, having ‘mixed evidence’ if only one reference benchmark is achieved, and ‘likely’ if both reference benchmarks are not achieved. We recommend combining ratings with other relevant data in a weight-of-evidence approach to assess if sediment impairs aquatic life.

Introduction

Human activities can increase sediment delivery to streams, changing the composition, distribution, and abundance of stream aquatic life (Wood & Armitage, 1997; EPA, 2006). To manage undesirable ecological effects of anthropogenic sediment, resource managers in the United States often compare stream surface fine sediment levels to a benchmark representing desired conditions (Dodds et al., 2010; Hawkins, Olson & Hill, 2010). Surface fine sediment is commonly measured as the percentage of sand and smaller particles (<2 mm diameter) using Wolman pebble count methods (Wolman, 1954), though other fractions and methods are also used. At the state and regional scale, government agencies use surface fines benchmarks to estimate the percentage of stream length with elevated sediment levels and inform management decisions (Van Sickle et al., 2006; Larson et al., 2019; Miller et al., 2021). At the subwatershed or reach scale, managers use surface fines benchmarks to assess if sediment is a stressor impairing aquatic life (EPA, 2006; Jessup et al., 2014). The Clean Water Act (CWA) requires states to identify stressor(s) impairing aquatic life and develop a restoration plan. Land management agencies also apply sediment benchmarks at the reach or subwatershed scale to identify stressors and evaluate the effectiveness of land management actions.

Few U.S. states have numeric surface fines water quality standards (WQS) under the CWA (EPA, 2006), so managers often use literature benchmarks or local data to develop application-specific benchmarks. Surface fines benchmarks have been developed using reference and stressor-response approaches. Reference benchmarks are indicator values associated with limited human landscape disturbance (Stoddard et al., 2006; Hawkins, Olson & Hill, 2010). Surface fines reference benchmarks have typically been defined as the 70th or 75th percentile surface fine sediment value for groupings of reference sites, with groupings selected to reduce fines variability (EPA, 2006; Jessup et al., 2014; Larson et al., 2019; EPA, 2020; Miller et al., 2021). A few studies have developed empirical models predicting site-specific surface fine sediment reference benchmarks based on continuous environmental gradients, but such models often have low r2 and large site prediction errors (Hawkins, Olson & Hill, 2010; Grangeon et al., 2023). Stressor-response benchmarks define stressor thresholds for a specific biological response of interest. They have been developed for surface fines levels associated with the initial onset of macroinvertebrate indicator changes (Burdon, McIntosh & Harding, 2013; Bryce, Lomnicky & Kaufmann, 2010; Relyea, Minshall & Danehy, 2012) and for levels associated with certain departure of macroinvertebrate indicators from reference condition (Jessup et al., 2014). Stressor-response benchmarks can be inferred from stressor-response plots (Relyea, Minshall & Danehy, 2012), based on quantile regression or changepoint analysis (Bryce, Lomnicky & Kaufmann, 2010; Burdon, McIntosh & Harding, 2013; Jessup et al., 2014), and based on logistic regression predicting the probability of a response across increasing substrate sediment levels (EPA, 2006). Some studies have developed several candidate benchmarks using multiple approaches to confirm stressor-response relationships and then have selected a benchmark that meets management goals (Bryce, Lomnicky & Kaufmann, 2010; Jessup et al., 2014).

In Idaho, state WQS do not include numeric criteria for stream bed sediment. Idaho WQS include a narrative sediment criterion requiring that sediment does not “impair designated beneficial uses” of water (IDAPA 58.01.02.200.08). Idaho WQS define full support of aquatic life beneficial uses as “where no biological group such as fish, macroinvertebrates, or algae has been modified significantly beyond the natural range of the reference streams or conditions approved by the Director in consultation with the appropriate basin advisory group” (IDAPA 58.01.02.010.42). Based on these requirements, the Idaho Department of Environmental Quality (IDEQ) established guidance for applying the narrative criterion (Idaho Department of Environmental Quality (IDEQ), 2016). Identifying narrative sediment criterion violations requires multiple lines of evidence, including (i) evidence of at least one human sediment source in the watershed, (ii) a potential transport pathway delivering anthropogenic sediment to a water body, and (iii) at least two lines of evidence for a measurable adverse effect of sediment on a beneficial use (Idaho Department of Environmental Quality (IDEQ), 2016).

The objective of this study was to develop a method for assessing if excess stream surface fine sediment likely alters macroinvertebrate community composition relative to reference conditions in Idaho wadable (1st–4th order) streams at the sample event scale. Our approach was designed to meet the desire for two lines of evidence for a measurable sediment-induced effect established in IDEQ guidance (Idaho Department of Environmental Quality (IDEQ), 2016). We developed reference benchmarks for percent surface fine sediment <2 mm and the macroinvertebrate fine sediment biotic index (FSBI) (Relyea, Minshall & Danehy, 2012), and stressor-response benchmarks indicating fines levels associated with a marginal (50%) and high (75%) probability that FSBI is worse than reference. We then applied benchmarks in a simple framework rating a sediment-induced macroinvertebrate community composition change as ‘likely’ when both surface fines and FSBI benchmarks are not achieved. We confirmed framework ecological relevance by examining associations between framework predictions and macroinvertebrate condition using data from multiple macroinvertebrate indicators and datasets1 .

Materials and Methods

BURP data

Benchmarks were developed using IDEQ Beneficial Use Reconnaissance Program (BURP) data. BURP is a bioassessment program that measures multiple stream physical, macroinvertebrate, and fish indicators in wadable 1st–4th order streams. IDEQ has collected BURP data in ~200 Idaho stream reaches during July-August each year in most years since 1993. BURP monitoring generally excludes federally designated wilderness areas, non-perennial streams, lake outlets, and reaches recently impacted by beaver activity or wildfire. Generally, BURP uses a targeted non-random sample design where crews select reaches representative of the associated stream assessment unit, the spatial scale used by IDEQ for CWA beneficial use support assessments. However, in several years an additional population of sites was also selected using a probability survey design for statewide data summaries. Data from both targeted and probability sampling were included in analyses, as described further below. BURP does not include repeat sampling of the same reach across years, but in some cases, sampling is spatially clustered across years and a single stream segment (National Hydrography Dataset Plus version 2 comid) was sampled multiple times.

All BURP data used in analyses were collected following protocols described in the Beneficial Use Reconnaissance Program Field Manual for Streams (Idaho Department of Environmental Quality (IDEQ), 2017). During each sample event field crews delineated a reach with length 30 times bankfull width or a minimum of 100 m. Macroinvertebrate samples were a composite of three samples, one from each of three separate riffle habitats within the reach. If three riffle habitats were not present, one or more samples were collected in a run. Samples were collected with a 500 µm mesh Hess sampler and composited and preserved with ≥70% ethanol in the field. Samples were identified to the lowest practical taxonomic level (typically to genus or species) and enumerated by EcoAnalysts Inc. following methods described in Richards et al. (2018). The same laboratory and methods were used across years. Substrate particle size was measured using modified Wolman pebble count methods (Wolman, 1954). Crews established three bankfull width transects, with each transect located one meter upstream of a macroinvertebrate sample location. Pebble counts were always conducted after macroinvertebrate samples to avoid disturbing macroinvertebrates during pebble counts. Within each transect at least 50 particles were selected at approximately equidistant intervals (heel to toe or one pace distance, etc.) and particle intermediate axis was measured and assigned to one of 11 size classes. Particles within both the wetted and non-wetted portion of the bankfull channel were included in counts. All particles within bankfull collected across the three transects were used to calculate percent surface fine sediment within the <2.5 mm size class. We assumed this size class represents sand and smaller particles and is functionally equivalent to surface fines measured as <2 mm. Hereafter we refer to percent surface fine sediment <2.5 mm as ‘SF’, shorthand for ‘sand and fines.’

Macroinvertebrate data were used to calculate fine sediment biotic index (FSBI) (Relyea, Minshall & Danehy, 2012) scores. FSBI is an index for assessing surface fine sediment impacts on macroinvertebrate community composition in northwest US streams. Relyea, Minshall & Danehy (2012) used macroinvertebrate data from multiple monitoring programs in the Pacific Northwest, including BURP data, to identify 206 taxa that are common in the Pacific Northwest (occurring in ≥2% of streams in their database). They assigned each taxon a sediment sensitivity score based on the percent fines <2 mm associated with the taxa’s 75th percentile occurrence. FSBI scores for a sample are then calculated as the sum of taxa-level scores (Relyea, Minshall & Danehy, 2012). FSBI is thus based on taxa occurrence only. FSBI scores generally range from 0–350; high scores indicate many sediment-sensitive taxa are present and therefore elevated surface fine sediment levels likely have not impacted macroinvertebrate community composition. Low FSBI scores indicate few sediment-sensitive taxa are present. FSBI has been applied in state-wide stream condition assessments in Washington State (Larson et al., 2019) and for assessing effects of streambed instability on macroinvertebrates (Kusnierz, Holbrook & Feldman, 2015). FSBI was used rather than other macroinvertebrate indicators because FSBI is sediment-specific and is correlated with more generalized macroinvertebrate indicators such as ephemeroptera, plecoptera, and tricoptera (EPT) taxa richness (Larson et al., 2019). In preliminary analyses there also was a stronger stressor-response relationship between SF and FSBI than with EPT taxa richness.

The ‘reference’ and ‘stress’ site definitions and datasets used by IDEQ to develop BURP bioassessment metrics were used for analyses (Jessup, 2011; Jessup & Pappani, 2015; Idaho Department of Environmental Quality (IDEQ), 2016). These reference and stress site definitions are routinely applied by IDEQ for CWA applications in Idaho (Idaho Department of Environmental Quality (IDEQ), 2016). BURP reference and stress sites were defined as the 10% least and 10% most-disturbed 1998–2007 BURP sites within each of three site classes, based on nine landscape indicators of upstream human landscape disturbance (Jessup & Pappani, 2015). Site classes are ecoregion groupings that help explain macroinvertebrate community structure variability within reference sites (Fig. 1). They were defined by applying non-metric multidimensional scaling and principal components analysis to Idaho macroinvertebrate data within BURP reference sites (Jessup, 2011; Jessup & Pappani, 2015). Site classes are described further in supplemental information. In cases where there were multiple BURP reference or stress sites within a stream segment (comid) across years, one reference or stress site per comid was randomly selected for use in analyses. Summary statistics for BURP reference and stress datasets used for analyses are in Table 1 for SF and Table 2 for FSBI.

Figure 1 Locations of site classes and stream reaches used to develop reference benchmarks (BURP data), stressor-response benchmarks (BURP data), and evaluate framework performance (PIBO and AIM data).

PPBV = plains, plateaus, and broad valleys.

Table 1 SF summary statistics for BURP reference and stress datasets used to develop benchmarks.

Order	Type	N	Min	Max	Mean	Median	
1	Reference	74	3.7	77.5	30.8	28.8	
Stress	38	2.5	100	48.6	48.3	
2	Reference	137	0.34	88.1	21.7	18.3	
Stress	43	2.55	100	44.4	39.4	
3	Reference	65	2.53	72.8	19.5	16.3	
Stress	34	0.58	100	46.0	35.1	
4	Reference	18	2.58	31.4	13.5	11.5	
Stress	17	0.47	99.0	36.7	13.0	

Table 2 FSBI summary statistics for BURP reference and stress datasets used to develop benchmarks.

Site class	Type	N	Min	Max	Mean	Median	
Foothills	Reference	26	20	195	108	102	
Stress	16	10	145	50.9	40.0	
Mountains	Reference	184	35	310	175	178	
Stress	40	0	250	115	120	
PPBV	Reference	31	0	180	58.9	50	
Stress	55	0	95	17.2	5	

Reference benchmarks

Reach-specific SF reference benchmarks (SFref) were developed based on relationships between SF and bankfull width within each stream order using data from BURP reference sites (Fig. 2). Strahler stream order used for benchmark development was based on National Hydrography Dataset Plus version 2 1:100,000 scale hydrography used by Idaho for CWA applications. Bankfull width and order both had strong correlations with SF (see supplemental information). SF decreased with order within 1st–4th order streams, and order appeared to be a reasonable categorical proxy for stream power and several other physical variables correlated with SF. Relyea, Minshall & Danehy (2012) observed surface fines decreased with stream order using a larger Northwest U.S. dataset. Miller et al. (2021) also observed strong relationships between bankfull width and SF in Idaho streams, and divided streams into groupings based on bankfull width to calculate SF reference benchmarks. Quantile regression was used to define SFref, the 75th percentile value expected among reference streams based on reach order and bankfull width (Fig. 2). All BURP sites that met reference criteria, including sites sampled using a targeted (N = 277) and probability survey design (N = 17) were used in regressions. We used quantile regression because previous studies indicated trying to predict site-specific reference fines condition exactly can result in order of magnitude or larger site-specific prediction errors (Hawkins, Olson & Hill, 2010; Grangeon et al., 2023). Local geomorphic features, within-reach processes, and local hill slope processes can have significant impacts on reach bed sediment conditions but can be challenging to capture in models (Hawkins, Olson & Hill, 2010; Snyder et al., 2013; Keestra et al., 2018). Using quantile regression allowed us to acknowledge this local variability. The 75th percentile was selected to represent an upper bound SF value expected under reference conditions. This approach is conservative (protective) because 25% of reference sites have SF values exceeding this threshold. The ‘quantreg’ R package (version 5.97) was used for regressions (Koenker, 2023). FSBI reference benchmarks (FSBIref) were calculated as the 25th percentile FSBI value within each BURP site class described above.

Figure 2 Relationship between bankfull width and % fines <2.5 mm by stream order.

The line is the 75th percentile quantile regression line for reference sites (blue circles). Grey triangles show BURP stress site data for comparison.

Stressor-response benchmarks

Logistic regression was used to predict SF benchmarks with a 50% (SR50) and 75% (SR75) probability that FSBI would be worse (less) than FSBIref. Unlike reference benchmarks, stressor response benchmarks were not reach-specific. Separate logistic regressions and benchmarks were developed for each stream order/site class combination. Regressions used all BURP data collected 1998–2021 (reference, non-reference, stress collected using either targeted or probability survey design) with SF as the predictor variable and a binary value (1 or 0) indicating if FSBI was less than FSBIref as the response variable. Logistic regressions were implemented using the ‘glm’ function in base R. Regression model fit was documented using the model chi square statistic, odds ratio confidence intervals, and Hosmer-Lemeshow (HL) goodness-of-fit test (Hosmer et al., 1997; Hosmer, Lemeshow & Sturdivant, 2013). The chi square statistic quantifies the difference between deviance associated with a null model with an intercept only and model deviance. A large and statistically significant (<0.05) chi square statistic provides evidence to reject the null hypothesis that the model does not perform better than random chance. The model odds ratio indicates how the odds of classifying a reach as having FSBI worse than reference increases for a 1% increase in SF. When lower (2.5%) and upper (97.5%) confidence intervals for model odds ratio values both exceed one, this provides evidence that increasing SF reduces FSBI. The HL goodness-of-fit test describes the level of agreement between observed and predicted outcomes by decile of predicted probability. The HL null hypothesis is that the model provides a good fit. Low HL test p values provide evidence for rejecting the null hypothesis and concluding model fit is poor. Statistics and logistic regression curve plots were used in a weight of evidence approach to evaluate whether developed regression models were adequate for purposes of defining stressor-response benchmarks.

Relative risk

Relative risk (RR) (Van Sickle et al., 2006) was calculated to describe the strength of association between poor SF condition and poor macroinvertebrate condition. RR was the ratio of two conditional probabilities: the probability that a macroinvertebrate indicator benchmark was exceeded when a SF benchmark was exceeded, divided by the probability that a macroinvertebrate indicator benchmark was exceeded when a SF benchmark was not exceeded. RR was calculated for SFref and each stressor-response benchmark using data from two probability survey datasets. First, RR was calculating using data from 85 1st–4th order stream reaches sampled across Idaho 2013–2016 using BURP methods and a spatially balanced probability survey design (Idaho Department of Environmental Quality (IDEQ), 2018). Site inclusion probabilities were based on stream order and total state stream length within each order. These BURP probability survey data were not used in reference benchmark development but were used for stressor-response benchmarks. Second, RR was calculated using 43 stream reaches sampled across Idaho Bureau of Land Management (BLM) lands 2013–2016 using a probability survey design and BLM Aquatic Inventory and Monitoring (AIM) program protocols (Bureau of Land Management, 2021; Miller et al., 2021). Site inclusion probabilities were based on stream order categories (1st and 2nd order, 3rd and 4th order, and 5th + order) and stream linear extent within each category (Miller et al., 2021). AIM collects paired Wolman pebble and macroinvertebrate data within each reach but uses a different sample design and field methods from BURP. Compared to BURP, more transects are used per reach, more macroinvertebrate subsamples are collected, and different macroinvertebrate collection methods are used, among other differences. A table comparing monitoring programs is included in supplemental information. Calculating RR using AIM data tests if associations between SF benchmark exceedance and macroinvertebrate effects persist across different monitoring approaches.

RR was calculated using SF as the stressor indicator and FSBI as the response variable. SF and FSBI benchmark status was used to rate SF and FSBI as either ‘good’ or ‘poor’ for each sampled reach. In addition, RR was calculated using SF as the stressor indicator and other macroinvertebrate indicators as the response variable. For BURP data, IDEQ’s multi-metric macroinvertebrate index (SMI2) was used. For AIM data, an index describing the reach-specific ratio of observed to expected (O/E) macroinvertebrate taxa (Miller et al., 2021) was used. FSBI is an index based on taxa sediment tolerances, whereas SMI2 and O/E are generalized indicators of macroinvertebrate community condition that are not specific to any pollutant. O/E describes the proportion of macroinvertebrate taxa expected under reach-specific reference conditions that were observed in samples. SMI2 reflects the macroinvertebrate community expected under BURP reference condition based on multiple individual macroinvertebrate metrics (Idaho Department of Environmental Quality (IDEQ), 2016). Program-specific SMI2 (≥52–54 depending on site class) and O/E (≥0.63) benchmarks were used to identify good and poor macroinvertebrate condition. For each dataset, RR values and an associated 95% confidence interval were calculated using the ‘spsurvey’ R package (Dumelle et al., 2023). Calculations used site weights (i.e., inverse of site inclusion probability) to generate unbiased RR estimates. RR values with a 95% confidence interval >1 indicate a risk of poor macroinvertebrate indicator condition when the SF benchmark is exceeded. Higher RR values suggest a stronger association between stressor and response indicators.

Application framework

We developed a simple framework for applying SF and FSBI benchmarks to rate the strength of evidence for a sediment-induced change in macroinvertebrate community composition for each sample event with paired data. The framework rates a sediment effect as ‘unlikely’ if both SFref and FSBIref benchmarks are achieved, as having ‘mixed evidence’ where only one reference benchmark is achieved, and as ‘likely’ if both reference benchmarks are not achieved. The framework was designed to be consistent with the desire for at least two lines of evidence for a sediment effect when evaluating compliance with Idaho’s narrative sediment criterion (Idaho Department of Environmental Quality (IDEQ), 2016).

We evaluated the framework by applying it to BURP, AIM, and PacFish/InFish Biological Opinion (PIBO) monitoring program (Kershner et al., 2004; Roper, Saunders & Ojala, 2019; Saunders et al., 2022) data collected throughout Idaho. PIBO data were not included in RR calculations because the PIBO sample design was not suitable for RR calculations. BURP, AIM, and PIBO sample events where paired Wolman pebble and macroinvertebrate samples were collected were included in analyses. Because some AIM and PIBO sites were sampled multiple times across years, we randomly selected one sample event per AIM and PIBO site to reduce spatial sampling bias among data used for analyses. Datasets used for evaluation included 1998–2021 BURP data (N = 5,215 sites/events), 2013–2022 BLM AIM data (N = 299 sites/events), and 2004–2019 PIBO data (N = 665 sites/events). Notched boxplots were used to examine how macroinvertebrate indicator (FSBI, SMI2, O/E) distributions varied with benchmark status and framework prediction classes. Statistically significant differences in median macroinvertebrate indicator values between condition categories were inferred by examining overlap (or lack thereof) between boxplot notches representing median 95% confidence intervals. All data and R code used for analyses are available online (https://doi.org/10.17605/OSF.IO/3cZRP). DEQ BURP data used were queried from DEQ’s internal BURP database. AIM and PIBO data were provided by USFS and BLM staff.

Results

Quantile regression equations used to develop reach-specific SF reference benchmarks are shown in Table 3 and benchmark distributions are shown in Fig. 3. SFref benchmark distributions overlapped across stream orders, but SFref values and distribution peaks decreased as order increased (Fig. 3). FSBI reference benchmarks were 76 for the Foothills site class, 140 for the mountains site class, and 20 for the PPBV site class. All SF stressor-response logistic regressions models and associated intercept and slope values were statistically significant (p < 0.05) and showed good model fit based on chi square and odds ratios (Table 4). All except two regression models (3rd and 4th order in the mountains site class) also showed good fit based on HL statistics. These two models had a significant HL p-value (p = 0.02–0.04) suggesting poor model fit but were retained because chi square values and odds ratios indicated good fit. Logistic regression curve plots are included in supplemental information. Order-specific stressor response SF benchmarks for 50% (SR50) and 75% (SR75) probability that FSBI is worse than reference ranged from 28–63% and 47–89% percent sand and fines respectively (Table 4). SR50 and SR75 benchmarks were similar between Mountains and Foothills site classes (Table 4). SR50 and SR75 benchmarks were consistently higher in the PPBV site class than in Mountains and Foothills classes (Table 4). For several stream order-site class combinations, SR50 benchmarks exceeded SFref distributions in Fig. 3, indicating there was <50% probability of FSBI worse than reference at SFref surface fines levels. SR50 benchmarks exceeded SFref distributions for 3rd–4th order streams in the mountains and foothills site classes, and for all orders in the PPBV site classes. In contrast, for 1st and 2nd order streams in the mountains and foothills site classes, SR50 overlapped with SFref distributions.

Table 3 Quantile regressions used to predict reference 75th percentile % fines <2.5 mm (SFref).

Order	Equation	
1	SFref = −3.22255 * BW + 47.73073	
2	SFref = −2.49803 * BW + 41.62932	
3	SFref = −1.24737 * BW + 34.97105	
4	SFref = −0.62823 * BW + 29.54360	
Note:

BW = bankfull width (m).

Figure 3 Distribution of reach-specific SF reference benchmarks by stream order predicted using quantile regressions.

For AIM and PIBO data, density distributions used one randomly selected sample event per site. N = 299 for BLM AIM, N = 5,215 for DEQ BURP, and N = 665 for PIBO.

Table 4 SF stressor-response benchmarks and logistic regression model statistics.

Site class	Order	N	SR50	SR75	Intercept (SE)	% fines B (SE)	Model X2	Odds ratio CI	HL p-value	
Mountains	1	175	34	58	−1.45 (0.2)	0.04 (0.006)	64.7	[1.03–1.05]	0.09	
2	392	37	55	2.18 (0.13)	0.06 (0.005)	209	[1.05–1.07]	0.12	
3	315	42	71	−1.54 (0.13)	0.04 (0.006)	39.4	[1.02–1.05]	0.04	
4	122	31	53	−1.07 (0.29)	0.04 (0.02)	4.03	[1.04–1.08]	0.02	
Foothills	1	503	35	52	−2.1 (0.43)	0.06 (0.01)	48.6	[1.04–1.07]	0.20	
2	1,320	41	61	−1.98 (0.25)	0.05 (0.006)	83.8	[1.04–1.07]	0.16	
3	803	34	56	−1.6 (0.22)	0.05 (0.007)	62.0	[1.03–1.06]	0.14	
4	158	28	47	−1.5 (0.32)	0.05 (0.01)	19.9	[1.03–1.09]	0.12	
PPBV	1	202	57	73	−3.75 (0.5)	0.07 (0.01)	73.3	[1.05–1.09]	0.3	
2	483	63	89	−2.5 (0.24)	0.04 (0.005)	73.3	[1.03–1.05]	0.13	
3	452	51	69	−2.9 (0.27)	0.06 (0.006)	143	[1.05–1.07]	0.24	
4	205	40	53	−3.05 (0.41)	0.08 (0.01)	76.5	[1.06–1.1]	0.29	

Relative risk results are shown in Table 5. RR lower 95% confidence intervals (CI) calculated using SFref and FSBIref were >1 for both BURP and AIM data, indicating surface fines >SFref was associated with FSBI worse than reference condition. When SFref was exceeded, FSBI was 1.8–2.5 times more likely to be worse than FSBIref than when SFref was achieved (Table 5). Poor SMI2 condition was 3.0 times more likely when SFref was exceeded than achieved. When using AIM O/E as the response variable, lower CI was 1 and poor O/E condition was 1.8 times more likely when SFref was exceeded than when achieved. SF50 and SF75 generally had weaker associations with macroinvertebrate condition than SFref. Lower CIs were <1 for four of eight cases examined (Table 5).

Table 5 Relative risk estimates and 95% confidence intervals for benchmarks.

Benchmark	RR (SMI2)	RR (O/E)	RR (FSBIref)	
	BURP	BLM AIM	BURP	BLM AIM	
SFref	3.0 (1.3–6.7)	1.8 (1.0–3.2)	2.5 (1.5–4.2)	1.8 (1.1–2.9)	
SR50	2.2 (0.9–5.3)	1.6 (0.9–3.1)	2.2 (1.3–3.9)	2.3 (1.5–3.4)	
SR75	1.7 (0.4–7.1)	2.2 (1.8–2.8)	0.8 (0.2–3.2)	2.0 (1.4–2.7)	

Boxplots indicated better (higher) FSBI scores were associated with fines <SFref across BURP, AIM, and PIBO data (Fig. 4). Notches on boxplots representing median FSBI 95% confidence intervals did not overlap between good (<SFref) and poor (>SFref) SF condition, indicating median FSBI values were significantly greater when SFref was achieved. When SFref was achieved, the 25th percentile FSBI score was better than or very close to FSBIref across all program/site class combinations, except in the mountains site class for BLM data (Fig. 4). Median O/E values were also significantly higher when SFref was achieved for AIM and PIBO data (Fig. 5). O/E values increased with FSBI (Fig. 5).

Figure 4 Relationship between SF benchmark achievement status and FSBI for BLM AIM, DEQ BURP, and PIBO data.

The dashed horizontal line is the FSBI reference benchmark. Box lower and upper ends are 25th and 75th percentile, the horizontal line is the median, notches indicate median 95% confidence intervals, whiskers extend to 1.5 times the interquartile range (IQR), and circles are data points 1.5 times the IQR.

Figure 5 For PIBO and AIM data, relationship between SF benchmark status and O/E (A), framework prediction and O/E (B), and between O/E and FSBI score and FSBIref status (C).

The dashed horizontal lines are the program-specific O/E benchmarks.

Assessment framework predictions also showed strong associations with generalized macroinvertebrate indicators (Figs. 5, 6). For PIBO and AIM data, reaches where the framework rated a sediment effect as ‘likely’ had significantly lower median O/E values than those where an effect was rated as having ‘mixed evidence’ or ‘unlikely’ (Fig. 5). Median O/E values did not achieve program O/E benchmarks in reaches with a ‘likely’ sediment effect and were better than program O/E benchmarks in ‘mixed evidence’ and ‘effect unlikely’ reaches (Fig. 5). Similar patterns were observed for BURP SMI2 data (Fig. 6).

Figure 6 Relationship between SF benchmark achievement status and SMI2 for DEQ BURP data.

The dashed horizontal line is the SMI2 benchmark. See Fig. 4 caption for boxplot descriptions.

Discussion

This study developed benchmarks and a simple framework for evaluating if excess surface fine sediment <2 mm may alter macroinvertebrate community composition changes at the reach sample event scale in Idaho 1st–4th order streams. Relative risk calculations and box plots indicated exceeding SFref was strongly associated with altered macroinvertebrate community condition measured using both the sediment specific FSBI and two generalized macroinvertebrate indicators (SMI2, O/E). Associations persisted across multiple datasets using differing sample design and field methods. This indicates our benchmarks and framework are ecologically relevant and potentially useful when applied to multiple monitoring programs in Idaho.

To our knowledge, only one previous study estimated surface fines benchmarks specifically for Idaho streams. Miller et al. (2021) defined reference benchmarks for surface percent fines <2 mm for Idaho using data from 226 Western U.S. reference reaches sampled 2000–2009 across multiple EPA regional bioassessment monitoring programs. They defined benchmarks for small (≤10 m bankfull width) and large (>10 m) streams within each of Northern Rockies and Northern Xeric Basins ecoregions and used the 70th percentile of reference sites. Their reference benchmarks ranged from 15–45% and overlapped with ours. Their benchmarks were lower for large streams than small streams, consistent with the pattern here (Fig. 2). EPA defined surface fines <2 mm reference benchmarks of 15% in the Northern Rockies and Pacific Mountains region for the National River and Streams Assessment (EPA, 2020), and 16.19–22.52% for wadable streams in the interior Columbia River Basin (EPA, 2007). For New Mexico streams, Jessup et al. (2014) calculated 75th percentile fines <2 mm benchmarks as 20.6–74.3% for various ecoregional groupings. A table summarizing literature-reported surface fines benchmarks is included in supplemental information. To our knowledge, this is the first study to develop FSBI reference and stressor-response benchmarks.

In relative risk analyses, we observed stronger associations between SFref and macroinvertebrate indicators than for stressor-response benchmarks using BURP data. RR values were similar for reference and stressor-response benchmarks using BLM AIM data (Table 5). Considering there was <50% probability of FSBI worse than reference when SFref was exceeded for several site class/order combinations (Fig. 3, Table 4), similar or higher RR values for reference benchmarks than stressor-response benchmarks may at first seem surprising. However, reference benchmarks likely had higher RR values than stressor-response benchmarks because of the wedge-shaped relationship between fines and FSBI (Fig. 7). In our study and in others using different macroinvertebrate indicators, macroinvertebrates typically show a nonlinear wedge-shaped response to increasing surface fines, with large macroinvertebrate indicator variability at low sediment levels and lower indicator variability and values as sediment levels increase (Bryce, Lomnicky & Kaufmann, 2010; Relyea, Minshall & Danehy, 2012; Jessup et al., 2014). This suggests surface fines are only one factor affecting macroinvertebrate indicators at low levels and their relative importance increases as surface fines levels increase. When using higher SF benchmarks such as SF50 and SF75, false negative macroinvertebrate responses increase, reducing RR values.

Figure 7 Surface fine sediment, FSBI, and framework predictions for BURP data by BURP siteclass and Strahler stream order.

The dashed horizontal line is the FSBI reference benchmark.

With a wedge-shaped response pattern, there is also potential for false positive macroinvertebrate responses at low sediment levels due to high macroinvertebrate response variability (Fig. 7). False positive rates were estimated as the percentage of reference sites where benchmarks were not achieved, or the framework predicted a sediment effect. When applied to sites that meet PIBO reference criteria (in wilderness or having no mining and minimal grazing, timber harvest and road density, see supplemental information) (N = 123), false positive rates were 17% for SFref and 41% for FSBIref, but 14% for framework predictions. When applied to BURP reference sites (N = 309), false positive rates were 27% for SFref and 21% for FSBIref, but 8% for framework predictions. By combining two lines of evidence, the framework reduces false positive rates. These patterns highlight that understanding the form of the stressor-response relationship for an indicator and benchmark performance is critical when selecting assessment protocols intended to manage a biological outcome.

Most studies calculating RR define three sediment condition categories–‘good’ (sediment < reference benchmark), ‘poor’ (sediment > poor condition benchmark) and ‘fair’ (in between) (Van Sickle et al., 2006; Larson et al., 2019; Kaufmann et al., 2022). Those studies calculated RR based on macroinvertebrate responses at ‘good’ and ‘poor’ sites, ignoring ‘fair’ reaches where stressor status was not clear. RR calculated this way tests for macroinvertebrate responses that occur when there is a relatively large magnitude increase in stressor levels from ‘good’ to ‘poor’. In contrast, we did not use ‘fair’ stressor and response condition categories. Our RR calculations test for the conditional probability that any SF increase above reference yields FSBI worse than reference. This approach is more protective but may reduce RR values relative to a 3-category approach. However, our RR values were comparable to those in other studies. Kaufmann et al. (2022) evaluated macroinvertebrate multimetric index responses to relative bed stability (RBS) for Western U.S. streams sampled through the EPA National River and Streams Assessment and reported RR = 2.64. Van Sickle et al. (2006) reported RR = 1.75 for macroinvertebrate index of biotic integrity (IBI) response to RBS in dataset from the Mid-Atlantic U.S. However, Larson et al. (2019) reported RR = 4 for a macroinvertebrate IBI response to % fines <2 mm for a Washington State probability survey dataset. Larson et al. (2019) used SF values > 25.5 to indicate poor condition, and <15.5 to indicate good condition. SF benchmark RR values may be higher in Larson et al. (2019) because they used a ‘fair’ condition category and defined poor conditions as SF values 10% greater than reference conditions, whereas our RR calculations assumed any magnitude SFref exceedance represented poor conditions. Landscape and geologic differences between Washington and Idaho may also contribute to RR differences between the two studies.

Our results demonstrate associations between SF and macroinvertebrate condition but cannot definitively confirm causation within a sampled reach. Degraded macroinvertebrate community condition clearly co-occurs with elevated SF levels. Relationships between SF and FSBI (Fig. 7) demonstrate that the number of sediment-sensitive macroinvertebrate taxa decreases as SF levels increase. This relationship was the basis for FSBI construction (Relyea, Minshall & Danehy, 2012). Boxplots also demonstrate associations between increasing SF and other macroinvertebrate condition measures (Figs. 5, 6). Relationships in Fig. 7 are also consistent with laboratory and mesocosm experiments documenting reduced density and richness of EPT taxa and other sediment-sensitive taxa in response to increasing sediment surface cover and fine sediment (Molinos & Donohue, 2009; Wagenhoff, Townsent & Matthaei, 2012; Conroy et al., 2016; Conroy et al., 2018). Definitively confirming causation in any individual sample reach would likely require dose-response experiments representative of reach conditions. Other approaches to rule out other potential causes, such as applying causal analysis decision frameworks (Seuter, 2007), structural equation modeling (Fergus et al., 2023), or Bayesian path analysis (Irvine et al., 2015) could also be used. Our framework predictions describe the strength of evidence for a sediment-induced macroinvertebrate community composition change. As discussed below, we recommend using framework predictions as only one line of evidence for stressor identification.

Application considerations

The benchmarks and framework developed here could be used to assist stressor identification. The CWA requires states to identify stressor(s) impairing aquatic life and develop a restoration plan to reduce stressor levels. When macroinvertebrate samples suggest the community is not within the “natural range of reference streams or conditions” and thus does not meet the definition of full support of aquatic life use in Idaho WQS (IDAPA 58.01.02.010.42), methods to diagnose the specific stressor(s) causing impairment are needed. Our benchmarks and framework could be used to assist stressor identification but should not be used as the only evidence diagnosing sediment impairment. BURP, AIM, and PIBO bioassessment programs all collect paired Wolman pebble and macroinvertebrate data, but also measure many other parameters relevant to assessing sediment-induced macroinvertebrate effects. The parameters measured differ across programs but include various stream physical habitat integrity parameters and generalized macroinvertebrate metrics. These other data should also be considered when diagnosing sediment impairments. Our framework also does not provide information on potential human sources or transport pathways, which are needed to assess compliance with Idaho’s narrative sediment criterion (Idaho Department of Environmental Quality (IDEQ), 2016). We recommend assessors combine benchmark and framework predictions with other lines of evidence in a weight of evidence approach, and explicitly articulate the lines of evidence and rationale for sediment assessment decisions.

When sediment has been confirmed as a stressor, the CWA requires states to define sediment targets to restore support of aquatic life use. Our benchmarks could be used as sediment restoration targets in streams where macroinvertebrate effects through substrate exposure pathways are the primary concern. Our approach does not address other sediment exposure pathways or ecological effects. Idaho WQS include numeric turbidity criteria to protect aquatic life from water column suspended sediment exposure pathways (IDAPA 58.01.02.250.02.3.e, IDAPA 58.01.02.401.02a). IDEQ guidance for selecting sediment targets also includes suggestions for suspended sediment concentration targets and substrate (surface plus subsurface) sediment targets based on streambed core samples for protecting salmonid spawning habitat (Idaho Department of Environmental Quality (IDEQ), 2003). IDEQ sediment target guidance did not include suggestions for surface fine sediment targets. This study represents one potential approach for selecting surface fine sediment targets.

Our benchmarks and framework predictions are specific to a single sample event within a single stream reach. Assessors conducting stressor identification analyses will likely face data from multiple sampled reaches within an area of interest or sample reaches with multiple sample events across years. Appropriate methods for addressing temporal variability or reconciling conflicting outcome predictions within or across reaches are application-specific and beyond the scope of this study. As described above, we recommend considering framework predictions along with other relevant lines of evidence relevant to a sampled reach. What lines of evidence are available will vary geographically, so appropriate decision rules will also. Management goals, such as the biological endpoint (macroinvertebrates, fish, endangered species, etc.) targeted often varies geographically, affecting decision rules needed.

Our SF reference benchmarks are based on channel bankfull width and implicitly assume reach bankfull width is within reference condition. In cases where human activities have increased bankfull width, we still expect the benchmark and framework to identify sediment-induced macroinvertebrate community changes. SF reference benchmarks decrease as bankfull width increases (Fig. 2) because in reference streams SF decreases with stream power. Generally, when human activities increase channel width stream power decreases and substrate fine sediment increases. In such cases, measured SF would likely exceed our SF reference benchmarks. Patterns in Figs. 4–6 suggest relationships between reference benchmark exceedance and macroinvertebrate responses are robust. The benchmarks and framework are not a tool for identifying stream channel geomorphic changes from reference condition but may complement other tools for evaluating anthropogenic channel alterations.

Conclusions

This study developed SF and FSBI benchmarks and a framework for applying them that can be used to evaluate the strength of evidence for sediment-induced macroinvertebrate community composition change in Idaho streams. We confirmed framework ecological relevance by examining associations between framework predictions and macroinvertebrate condition using data from multiple macroinvertebrate indicators and datasets. We propose rating the strength of evidence for a surface fine sediment-induced macroinvertebrate community composition change at the sample event scale as ‘unlikely’ if both SF and FSBI reference benchmarks are achieved, having ‘mixed evidence’ if only one reference benchmark is achieved, and ‘likely’ if both reference benchmarks are not achieved. Ratings should be combined with other relevant data in a weight-of-evidence approach to assess if sediment impairs aquatic life.

Supplemental Information

Supplemental Information 1 Supplemental information.

Carl Saunders and Trip Armstrong provided PIBO data. Trip Armstrong, Logan Shank, and Jennifer Courtwright provided BLM AIM data. Tony Olsen (USEPA) provided guidance on relative risk calculations. Brady Johnson (IDEQ) conducted a quality assurance review of project R code. We thank Justin Jimenez, Eric Archer, Jennifer Courtwright, and Doug Peterson for feedback on earlier versions of this study.

Additional Information and Declarations

Competing Interests

Author Contributions

Data Availability

1 We previously published portions of this text in a preprint (https://doi.org/10.31223/X58M4M).

The authors declare that they have no competing interests. Jason Williams was employed at the Idaho Department of Environmental Quality when the research was performed and now is employed by The Cadmus Group. James Efta is employed by the United States Forest Service.

Jason Williams conceived and designed the experiments, analyzed the data, prepared figures and/or tables, authored or reviewed drafts of the article, and approved the final draft.

James Efta conceived and designed the experiments, authored or reviewed drafts of the article, and approved the final draft.

The following information was supplied regarding data availability:

The data and R code used for analyses are publicly available at the Open Science Framework: Williams, Jason. 2024. “A Method for Evaluating Sediment-Induced Macroinvertebrate Community Composition Changes in Idaho Streams.” OSF. June 17. DOI 10.17605/OSF.IO/3CZRP.

The DEQ BURP data used were queried from DEQ’s internal BURP database. AIM and PIBO data were provided by USFS and BLM staff.

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
