# Peer review of "A method for evaluating sediment-induced macroinvertebrate community composition changes in Idaho streams"

_PeerJ, doi:10.7717/peerj.18060_

## Round 0.1 · original submission · Minor Revisions

Dear Dr. Williams,

After this first review round, both reviewers believe there are only minor reviews to be made. Please implement the required changes and I believe the manuscript will be accepted for publication.

SIncerely,
Daniel Silva

Reviewer 1 ·

Basic reporting

Well written and thoroughly described. Literature references are appropriate and relevant and provide helpful background on the motivation for the study.

The abstract might grab more readers by including more motivation and background for the study. As written it immediately launches into what the study does – which is great but may narrow the audience. Including some of the background from the introduction in the abstract would help.

Experimental design

The authors apply a novel combined approach to address an environmental and management concern using sound logic, methodology, and a simple evaluation framework. This line of work is needed to guide the protection of aquatic resources from stressors with limited criteria.
The authors thoroughly document their method and describe the datasets they worked with such that others could replicate the work or apply similar approaches to other datasets with appropriate variables. I’m not sure if the raw data will be shared.

Validity of the findings

Figures and tables support the main conclusions as described in the results. Conclusions are well stated and the Discussion is nicely written describing the main findings and implications. I'm not sure if the underlying data will be provided.

Additional comments

L392-401 It would be helpful to explain the false positive rates in more detail. How are they estimated? How are they validated?

L417-419 Could you expand upon why RR is much higher in Larson et al. and what that may mean to the RR calculated in this study?

Figure 1 – it may be helpful to label each panel and provide n count of observations per dataset

Figure 5 Caption – make sure to identify each of the panels – part of text may have been cut off at the end.

·

Basic reporting

This is well-structured and well-written. I left some comments where I was initially uncertain of the authors' meaning, but the meaning became clear later in the paper. Citations seem relevant and complete.

Experimental design

Good. The later introduction of AIM and PIBO data made it somewhat confusing about which analyses used those data.

Validity of the findings

This is very relevant for Idaho and beyond, especially with the comparison among data sets and to other benchmarks.

Additional comments

Nice work! I could follow your reasoning and analysis, though some sections needed a second read to understand. This means that the information was there, but not always intuitive to me (not your problem). Hopefully your results will contribute to adoption of assessment tools in Idaho.

---

## Round 0.2 · accepted · Accept

Dear Dr. Williams,

I am pleased to accept your manuscript for publication in PeerJ. Congratulations!

Sincerely,
Daniel SIlva

·

Basic reporting

Revisions are good.

Experimental design

Revisions are good.

Validity of the findings

Revisions are good.

Additional comments

Revisions are good.